# Advancements in Portable Voltammetry: A Promising Approach for Iron Speciation Analysis

**DOI:** 10.3390/molecules28217404

**Published:** 2023-11-03

**Authors:** Paolo Inaudi, Ornella Abollino, Monica Argenziano, Mery Malandrino, Caterina Guiot, Stefano Bertinetti, Laura Favilli, Agnese Giacomino

**Affiliations:** 1Department of Drug Science and Technology, University of Torino, 10125 Torino, Italy; ornella.abollino@unito.it (O.A.); monica.argenziano@unito.it (M.A.); laura.favilli@unito.it (L.F.); agnese.giacomino@unito.it (A.G.); 2Department of Chemistry, University of Torino, 10125 Torino, Italy; mery.malandrino@unito.it (M.M.); stefano.bertinetti@unito.it (S.B.); 3Department of Neurosciences “Rita Levi Montalcini”, University of Torino, 10125 Torino, Italy; caterina.guiot@unito.it

**Keywords:** iron, speciation, voltammetry, water, antimony–bismuth film

## Abstract

Iron, a crucial element in our environment, plays a vital role in numerous natural processes. Understanding the presence and concentration of iron in the environment is very important as it impacts various aspects of our planet’s health. The on-site detection and speciation of iron are significant for several reasons. In this context, the present work aims to evaluate the applicability of voltammetry for the on-site determination of iron and its possible speciation using a portable voltammetric analyzer. Voltammetry offers the advantage of convenience and cost-effectiveness. For iron (III) determination, the modification of a glassy carbon electrode (GCE) with an antimony-bismuth film (SbBiFE) using the acetate buffer (pH = 4) as a supporting electrolyte was used. The technique adopted was Square Wave Adsoptive Cathodic Stripping Voltammetry (SW-AdCSV), and we used 1-(2-piridylazo)-2-naphthol (PAN) as the iron (III) ligand. Linearity, repeatability, detection limit, and accuracy were determined using synthetic solutions; then, a Standard Reference Material (SRM) of 1643f Trace Elements in Water (iron content: 93.44 ± 0.78 µg L^−1^) was used for validation measurements in the real matrix. the accuracy of this technique was found to be excellent since we obtained a recovery of 103.16%. The procedure was finally applied to real samples (tap, lake, and seawater), and the results obtained were compared via Inductively Coupled Plasma-Optical Emission Spectroscopy (ICP-OES). The amount of iron found was 207.8 ± 6.6 µg L^−1^ for tap water using voltammetry and 200.9 ± 1.5 µg L^−1^ with ICP-OES. For lake water, 171.7 ± 3.8 µg L^−1^, 169.8 ± 4.1 µg L^−1^, and 187.5 ± 5.7 µg L^−1^ were found using voltammetry in the lab both on-site and using ICP-OES, respectively. The results obtained demonstrate the excellent applicability of the proposed on-site voltammetric procedure for the determination of iron and its speciation in water.

## 1. Introduction

Iron has both positive and negative impacts on the environment, depending on its form, concentration, and distribution [1,2]. Iron is an essential micronutrient for living organisms, including humans, plants, and animals. As for humans, it plays a crucial role in various biological processes, such as oxygen transport, energy production, and enzyme function. Furthermore, it is a component of soil minerals and plays a vital role in maintaining soil fertility [3,4]. It affects soil structure, pH balance, and nutrient availability [1]. Adequate iron levels promote healthy plant growth and development. On the other hand, high concentrations of iron in water can lead to contamination. Iron-rich groundwater or surface water can lead to discoloration, presenting a reddish or brownish tint. This iron-contaminated water may be unsuitable for certain uses, such as drinking, irrigation, or industrial processes.

Iron is present in waters at concentrations that vary greatly among sea, lake, or drinking water. Soluble iron is found in seawater in very low concentrations (nanomolar or picomolar) due to some phenomena, such as particle scavenging, low solubility, and the effective removal caused by biological absorption. These processes are less present in other waters, such as lake water or drinking water, in which iron concentrations are higher [5,6,7,8].

Iron also has a relevant influence on human health: whereas it is an essential nutrient required for various physiological processes, its dysregulation can contribute to the pathogenesis of neurodegenerative disorders such as Alzheimer’s disease, Parkinson’s disease, and Huntington’s disease [9,10,11,12,13]. In a recent paper, Baringer et al. [14] emphasized the fundamental role that iron plays in human neurological health and, in this way, many studies have been carried out in order to understand its role and facilitate its determination as a marker of such diseases [10,15].

Electrochemical methods are widely used for iron determination and speciation due to their sensitivity, selectivity, and ability to provide real-time measurements [4,16]. Over the years, different voltammetric techniques and iron complexants that are able to enhance the selectivity and sensitivity of responses, have been tested. Differential pulse voltammetry (DPV), Cyclic Voltammetry (CV) and Square Wave Voltammetry (SWV) are the most commonly used techniques [17,18,19,20,21]. A wide range of complexing agents have been used, such as 1-nitroso-2-naphthol (NN) [22], 4,4′-[3-(2-pyridyl)1,2,4-triazine-5,6-diyl]bis(benzene sulfonic acid) disodium salt hydrate (ferrozine, FZ) [23], dihydroxynaphthalene (DHN) [24,25], 2-(5-bromo-2-pyridylazo)-5-diethylaminophenol (5-Br-PADAP) [26] and 2-(2-thiazolylazo)-p-cresol (TAC) [27]. Another fundamental aspect is the use of suitably modified working electrodes (WE); in the literature, there are some important examples, such as chemically modified carbon-paste electrodes (CMCPEs) [28,29], nafion-coated electrodes (NCE) [30], thick-film graphite containing electrode modified with calomel (TFGME) [31], boron-doped diamond (BDD) electrodes [32], bismuth-coated glassy carbon electrodes (BiFE) [33,34], gold and bismuth bimetallic nanoparticles decorated with l-cysteine-functionalized graphene oxide nanocomposites (Au-BiNPs/SH-GO) [35] and gold 2-mercaptosuccinic acid self-assembled monolayers (AuMSA SAM) [36]. In recent years, even more complex electrode modifications have been tested for simultaneous iron (II) and iron (III) detection using, e.g., nitrogen-doped carbon quantum dot silver nanoparticle β-cyclodextrin nanomaterials (N-CQDs/AgNPs/β-CD) [37]. The use of electrode-specific modifiers and complexants greatly helps increase the selectivity and sensitivity of these voltammetric methods.

An advantage of voltammetry is the availability of portable instruments that enable on-site measurements and are useful in various fields, such as clinical practice, quality control, and environmental monitoring.

The aim of this study was to develop a new portable voltammetric method for the on-site determination of iron (III) in different types of water. This work is focused on the use of a new antimony-bismuth film-modified glassy carbon electrode (SbBiFE-GCE) coupled with 1-(2-piridylazo)-2-naphthol (PAN) as a ligand for iron (III). All the analyses were carried out in triplicate with the use of a portable potentiostat and tested outside laboratories.

## 2. Results and Discussion

The aim of this paper was to develop a method that allowed iron (III) to be selectively analyzed when present in different matrices directly in the field.

The decision to modify the electrode with a film of Sb-Bi was made starting from the study of Segura et al. [33], where only Bi was used, integrating with other recent studies using Sb for the modification of WE.

The optimization of conditions for film formation was carried out by testing different concentrations of Bi and Sb and different deposition times. As for the potential of deposition and the supporting electrolyte, it was decided to use those reported in some papers regarding only the bismuth film [33,34]. The tests were carried out by evaluating the accuracy of determining a concentration of 5 µg L^−1^ of iron (III) in a synthetic solution (10 mL of 0.1 mol L^−1^ acetate buffer (pH 4) and 5 µmol L^−1^ PAN).

In particular, the conditions reported in Table 1 were tested.

In tests 2–3, using 300 and 200 mg L^−1^ of Bi and Sb, respectively, with a deposition time of 120 and 30 s, the film did not stick to the WE surface, and iron was not determined (n.d. in Table 1); after 300 s (test 1), the results were better, but the film was thicker, and the sensitivity (data not showed) was worse than the analog with lower concentrations of Bi and Sb (test 4) as well as the obtained recovery.

Tests 7–9 experienced similar problems as follows: with short deposition times, iron was not determinable. The best results were obtained with test number 4, with a percentage recovery of 95.3%.

All the preliminary tests were made first on synthetic solutions (Table 1) and then on a SRM 1643f using the SWAdSW parameters reported.

To obtain good results in terms of sensitivity, different concentrations of PAN were tested, namely 1, 5, and 10 µmol L^−1^. The concentration that permitted the best response was 5 µmol L^−1^ in the cell, confirming the findings by Segura et al. [33]; tests made at higher and lower concentrations (10 µmol L^−1^ and 1 µmol L^−1^) gave rise to a worsening performance for this method. In particular, with a higher concentration, we lost sensitivity (87.5% recovery), and with a lower concentration, iron was not displayed.

Tests were also made using a standard of iron (II), showing an absence in the signal and consequently certifying the selectivity of the method and of the complexant used for iron (III) in the experimental conditions adopted.

In the second step of this work, samples were analyzed for the total iron, as described in Section 2.2 and Section 2.3, after the addition of nitric acid in order to oxidize any iron (II) present.

The results obtained for total iron were also confirmed by another analytical technique, namely ICP-OES.

### 2.1. Synthetic Solutions

The first tests were carried out on synthetic solutions. The electrochemical cell was composed of 10 mL of a 0.1 mol L^−1^ acetate buffer (pH 4) and 5 µmol L^−1^ of PAN, to which subsequent aliquots of iron (III) were added.

Figure 1 reports voltammograms for the blank and four successive additions of iron (III). The analytical signal was the current intensity registered for the potential of −0.475 V.

The results obtained are shown in Table 2.

Reproducibility studies were performed intracell, intercell, in the laboratory and in the field with a relative % RSD of 2.8%, 3.0%, 4.1%, and 3.9%. In Figure 2, the voltammogram of three standard additions of iron (II) and a subsequent one of iron (III) demonstrates the selectivity of this method for iron (III).

The true iron concentration of the home-made standards used for the preparation of synthetic solutions was checked using ICP-OES for the analysis of the total iron present in the solution. The concentrations were computed by means of an external calibration obtained via dilution from a concentrated (1000 mg L^−1^) Sigma-Aldrich standard solution. The results obtained, as shown in Table 3, confirm the reliability of the iron (III) and iron (II) standards.

### 2.2. Certified Reference Material

After that, the accuracy of the method was tested on a Standard Reference Material (SRM) 1643f—Trace Elements in Water—with a total iron-certified concentration of 93.44 ± 0.78 µg L^−1^.

The iron (III) concentration in SRM was estimated using the standard addiction method, as reported before. The accuracy of this technique was found to be excellent since we found 96.40 ± 2.45 µg L^−1^ of iron with a recovery of 103.16%. The SRM 1643f contained HNO_3_ as a stabilizer; therefore, it was not surprising to find all the iron as iron (III).

Figure 3 shows the voltammogram for this test.

### 2.3. Water Samples

After optimizing and verifying the effectiveness of this method, it was tested on real samples of water (tap water, sea water, and lake water).

For lake water, the same sample was analyzed both in the laboratory and in the field to demonstrate the applicability of this method outside the laboratory.

Figure 4 and Figure 5 show the voltammograms of lake water and tap water, respectively.

For lake and tap water, the method responded excellently; in detail, the tests were carried out in triplicate, and a comparison with ICP-OES was also made. Before the analysis, nitric acid (final concentration 0.01 mol L^−1^) was added to the water samples. In this way, all the iron in the solution was present as iron (III), and a direct comparison was performed between voltametric and ICP-OES analysis.

The amount of iron found was 171.7 ± 3.8 µg L^−1^ using voltammetry in the first test and 187.5 ± 5.7 µg L^−1^ with ICP-OES. For tap water, 207.8 ± 6.6 µg L^−1^ and 200.9 ± 1.5 µg L^−1^ were found via voltammetry and ICP-OES, respectively. A T-test was conducted to assess whether the results obtained with the two techniques were significantly different. For tap water, there were no significant differences between the two data (*p*-value = 0.14). In the case of lake water, the values were statistically different (*p*-value = 0.03); this was probably due to the fact that being a more complex matrix, not all iron (II) was oxidized to iron (III). In fact, the value at the ICP-OES, which evaluates the total iron, turned out to be higher.

However, it was not possible to determine the concentration of iron in seawater, probably due to the fact that the concentration of iron is presumably lower than the LOQ of the proposed technique. As an example, seawater from the eastern Mediterranean Sea contains around 83 ng L^−1^ of iron [38]. In any case, standard additions are displayed, and the instrumental LOQ was 2.07 μg L^−1^.

This technique was found to be very accurate for water analysis even in the field; the lake water was also analyzed directly on-field, and the result was 169.8 ± 4.1 µg L^−1^ compared to the 171.7 ± 3.8 µg L^−1^ of the laboratory. Figure 6 shows a voltammogram of an on-site analysis of lake water. However, the greatest limitation of this procedure was the limited duration of the antimony and bismuth film. In fact, it was necessary to regenerate it every three samples in order not to lose sensitivity. The primary objective was to identify optimal conditions that could enhance the film’s longevity. Subsequently, it is anticipated that future investigations could encompass additional surface modifications of the glassy carbon electrode (GCE) to explore further enhancements.

The innovative nature of this technique concerns its capacity to facilitate direct analytical assessments in remote geographic areas, enabling the real-time monitoring of iron concentrations while avoiding the logistical complexities associated with sample collection and transportation, particularly in challenging environmental contexts.

The future development of this study will be testing the applicability of the procedure to the analysis of biological samples (such as plasma or serum). Even more challenging is the detection of iron in Cerebrospinal Fluid (CSF), ranging from about 30 in controls to 70 μg L^−1^ in AD patients [39]. The determination of iron in CSF samples is carried out by means of a Graphite Furnace Atomic Absorption Spectrometer (GF-AAS), which is a very precise but time-requiring method. For the voltammetric analysis of CSF, matrix effects have to be removed. The following approaches may be used: centrifuge the sample to separate solid particles or precipitates from the liquid phase, adjust the pH of the sample, and dilute the samples with the supporting electrolyte at least 1:10.

## 3. Material and Method

### 3.1. Instruments

Square Wave Adsorptive Cathodic Stripping Voltammetry (SW-AdCSV) was carried out with a Palmsens 4 portable potentiostat (PalmSens, Houten, The Netherlands). The height of the peaks was always taken in “automatic mode” by the software (PS trace 5.9). A classic electrochemical cell was equipped with a glassy carbon WE (GCE, 2 mm diameter, Metrohm, Herisau, Switzerland), a Ag/AgCl/KCl 3 mol L^−1^ reference electrode (RE), and an auxiliary platinum electrode. A mechanical stirrer (IKA-Topolino, Staufen, Germany) was connected to the PalmSens4 and powered by a portable battery for on-site analysis. A Cyberscan 2100 pH meter (Eutech Instruments srl, Thermo Fisher Scientific, Waltham, MA, USA) was used to adjust the solution’s pH, and the pH meter was calibrated daily with pH 4 and pH 7 buffers. An Inductively Coupled Plasma-Optical Emission Spectrometer (ICP-OES), in particular Perkin Elmer Optima 7000 (Perkin Elmer, Norwalk, CT, USA), was used for the measurement of total iron in the test analysis.

### 3.2. Reagents

All solutions were prepared with high-purity water (HPW-Milli-Q, Millipore, 18.2 MΩ cm, Milford, MA, USA). Antimony and bismuth standard solutions were obtained via dilution from concentrated (1000 mg L^−1^) Merck TraceCERT stock solutions (Merck, Darmstadt, Germany). Ammonium iron (II) sulfate hexahydrate and iron (III) nitrate nonahydrate (Sigma Aldrich-Merck, Darmstadt, Germany) were used for the preparation of iron (II) and iron (III) standards, respectively. Standard solutions were prepared for both analytes at a 1000 mg L^−1^ concentration and subsequently diluted to 1 mg L^−1^. Acetate buffers (0.1 mol L^−1^, pH 3.0–6.0) were prepared using potassium acetate and acetic acid (Sigma Aldrich-Merck, Darmstadt, Germany). A 0.001 mol L^−1^ solution of PAN (Sigma Aldrich-Merck, Darmstadt, Germany) was prepared in ethanol. Standard Reference Material (SRM) 1643f Trace Elements in Water (NIST, National Institute of Standards and Technology, Gaithersburg, MD, USA) with a total iron-certified concentration of 93.44 ± 0.78 µg L^−1^ were used for validation measurements.

### 3.3. Preparation of –SbBiFE-GCE

Before using the GCE was sequentially polished with three different alumina powders of different granulometry, namely 0.05, 0.3, and 1 µm (CH Instruments, Austin, TX, USA) on a microcloth pad (BASi, West Lafayette, IN, USA) and then rinsed with HPW and ethanol. This mechanical cleaning of the GCE was carried out daily.

After that, GCE was modified by the formation of a double film of bismuth and antimony in two successive steps. All parameters for film deposition and the amount of bismuth and antimony were optimized and are reported below.

The first step was the deposition of the antimony film; in particular, a cell containing 0.1 mol L^−1^ of the acetate buffer at pH 4.5 and 10 mL of the solution with an antimony concentration of 50 mg L^−1^ was prepared. A −1.0 V deposition potential was applied for 300 s.

The second step, analogously to the first one, allowed the subsequent deposition of bismuth onto the WE surface in the same supporting electrolyte but with a bismuth concentration equal to 100 mg L^−1^ in the 10 mL cell. The parameters for deposition were the same as those adopted for antimony.

### 3.4. Parameters for Iron Determination

For iron determination in a synthetic solution, a cell containing 10 mL of a 0.1 mol L^−1^ acetate buffer (pH 4) with an optimized and known concentration of PAN (5 µmol L^−1^) was used.

For the analysis of the certified sample, the cell consisted of 0.5 mL of the sample, 9.5 mL of the acetate buffer 0.1 mol L^−1^ (pH 4), and an amount of 50 µL of PAN (final concentration 5 µmol L^−1^ in the cell). The concentration was quantified with the standard addition method by adding two successive aliquots of 5 μg L^−1^ of iron (III). The test was carried out in triplicate.

Lake water was collected from Avigliana Lake (Avigliana, TO, Italy), while the sea water was sampled in Savona (Italy). The water samples were sampled using 1 L plastic bottles, stored in the fridge, and the analysis of the sample was carried out within 24 h of collection.

For these samples, analyses were carried out using a voltammetric cell containing 0.25 mL of the sample, 9.75 mL of the 0.1 mol L^−1^ acetate buffer (pH 4), and 5 µmol L^−1^ of PAN. Then, two standard additions of iron (III) (10 μg L^−1^ each) were made, and only two standard additions were performed on the samples due to the limited durability of the film.

The parameters optimized for iron (III) determination included deposition at −0.4 V for 180 s and a subsequent scan from −0.3 V to −0.9 with a frequency of 10 Hz and an amplitude of 0.025 V.

## 4. Conclusions

The proposed method has been shown to be excellent for the determination of iron (III) in water, using a method that is also applicable in the field.

The results obtained and the precision and accuracy of the optimized procedure make it possible to hypothesize its application in different areas. A future development of this method could be to increase the stability of Bi/Sb, thus reducing the frequency of its regeneration.

Overall, the time required for a complete voltammetric analysis, including film deposition, is 30 min; this value is acceptable, and a throughput of 25 samples for 8 h of operation is expected. This method is also suitable for speciation studies when considering that iron (II) is not detected using the proposed experimental conditions. This can allow for a potential increase in water analyses in cases of iron contamination or to investigate the role of this metal in biogeochemical cycles, even directly on site. This possibility could enable researchers to perform the rapid and immediate monitoring of iron concentrations, testing its suitability as a possible marker for the early detection of neurodegenerative diseases.

## Figures and Tables

**Figure 1 molecules-28-07404-f001:**
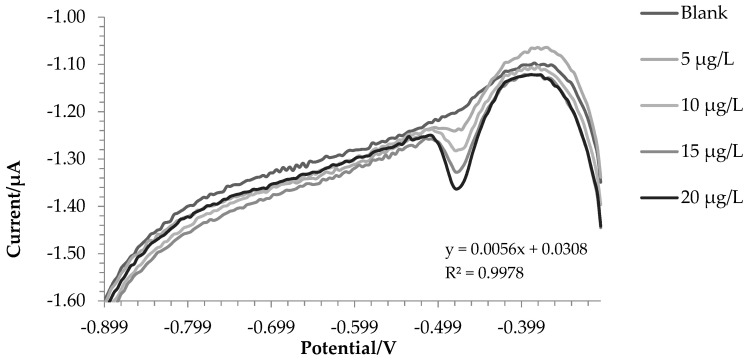
Voltammograms of the blank solution and successive additions of iron (III).

**Figure 2 molecules-28-07404-f002:**
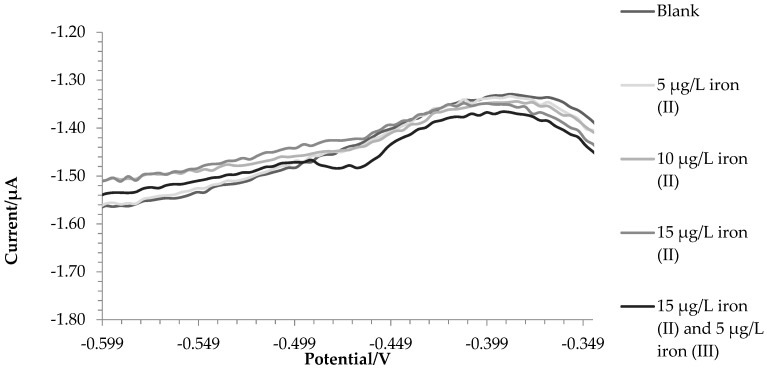
Voltammograms of the blank solution, successive additions of iron (II) and an addition of iron (III).

**Figure 3 molecules-28-07404-f003:**
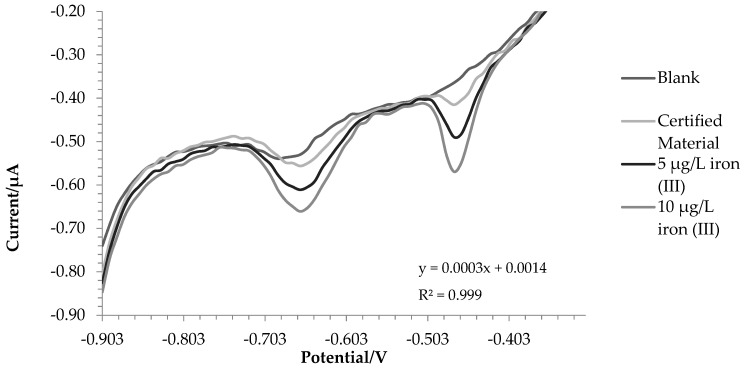
Voltammograms of SRM and two standard additions of Fe(III).

**Figure 4 molecules-28-07404-f004:**
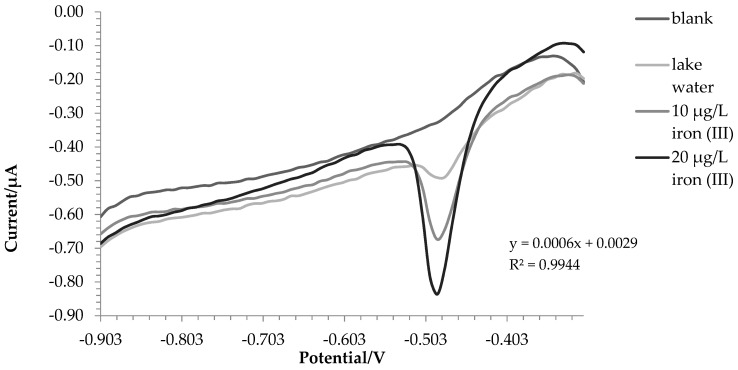
Voltammograms recorded for lake water analysis.

**Figure 5 molecules-28-07404-f005:**
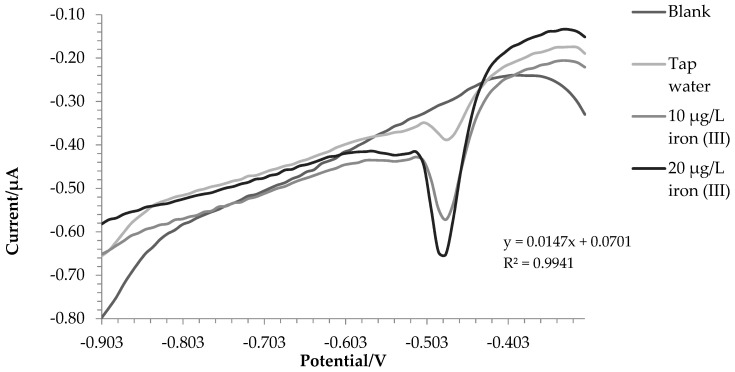
Voltammograms recorded for tap water analysis.

**Figure 6 molecules-28-07404-f006:**
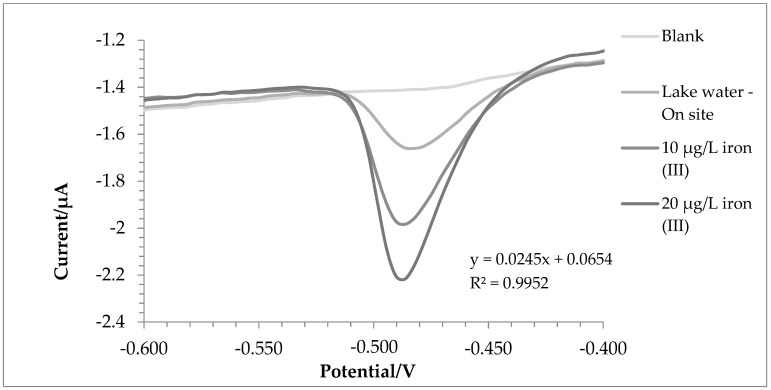
Voltammograms recorded for lake water for on-site analysis.

**Table 1 molecules-28-07404-t001:** Tests for best SbBiFE performance.

Test	Bi Concentration (mg L^−1^)	Sb Concentration (mg L^−1^)	Deposition Time (s)	Accuracy (% of Recovery)
1	300	200	300	76.7
2	300	200	120	n.d.
3	300	200	30	n.d.
4	100	50	300	95.3
5	100	50	120	89.8
6	100	50	30	88.7
7	20	10	300	80.5
8	20	10	120	n.d.
9	20	10	30	n.d.

**Table 2 molecules-28-07404-t002:** Figures for the merit of the developed method.

WE	Analyte	Repeatability (RSD %, *n* = 3)	Linearity	LOD * (μg L^−1^)	LOQ * (μg L^−1^)	Accuracy (% Recovery)
**SbBiFE**	Iron (III)	3.05	y = 0.056x + 0.0308 R^2^ = 0.9978	0.54	1.78	95.4

* Limit of Detection (LOD) and Limit of Quantification (LOQ) were estimated as 3 and 10 times the standard deviation of the blank (*n* = 3), respectively.

**Table 3 molecules-28-07404-t003:** Analysis of prepared iron standards vs. ICP-OES.

Analyte	Theoretical Concentration Prepared (μg L^−1^)	Results with ICP-OES (μg L^−1^)	Recovery (%)
Iron (II)	1000	1058	105.8
Iron (III)	1000	1034	103.4

## Data Availability

Data, associated metadata, and calculation tools are available from the corresponding author (paolo.inaudi@unito.it).

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
