# Peer review of "Advancements in Portable Voltammetry: A Promising Approach for Iron Speciation Analysis"

_molecules, 2023, doi:10.3390/molecules28217404_

Round 1

Reviewer 1 Report

Comments and Suggestions for Authors

(molecules-2671518)

The current work provides an electroanalytical approach for the on-site determination of iron (III) in different types of water using a new portable voltammetric method.

The manuscript is well-prepared, and easy to follow.

However, the value of the conducted experiments is rather negligible, as the sensor presented in the manuscript may be used for not more than three samples, which in Reviewer opinion make it rather useless and time-consuming.

In Reviewer opinion the Authors should add some short description connected with the novelty of presented approach, because now it seems that the manuscript doesn't bring anything innovative with it.

Another thing: what are the advantages of the proposed procedure in comparison to the others methodologies, known in literature?

Following remarks should be also taken into account:

1.      The novelty of the manuscript is rather limited. The use of Sb should be explained, why the Authors decided to modify the electrode surface together with Sb and Bi??

Maybe some kind of comparison, of the developed sensor should be performed, for example bare electrode vs Bi modified electrode vs Sb/Bi modified electrode. And based on such results it can be concluded does this research make sense.

2.      Page 5, what is the linear range of the proposed procedure?

3.      What about Figure 2? Is it not mentioned in the text, what is the conclusion about these data??

How the peak currents were measured?

4.      If the Authors mentioned that some of analyses were performed outside of the lab, then it is interesting to look at these data, if the signals differ somehow. Then Authors should add voltammograms for on situ analyses.

Based on the above I recommend the major revision of the manuscript.

Author Response

The current work provides an electroanalytical approach for the on-site determination of iron (III) in different types of water using a new portable voltammetric method. The manuscript is well-prepared, and easy to follow.

However, the value of the conducted experiments is rather negligible, as the sensor presented in the manuscript may be used for not more than three samples, which in Reviewer opinion make it rather useless and time-consuming. In Reviewer opinion the Authors should add some short description connected with the novelty of presented approach, because now it seems that the manuscript doesn't bring anything innovative with it.

We thank the Reviewer for his/her kind and helpful comments.

The main novelty of the method concerns the possibility of carrying out speciation studies for the determination of iron (III) directly in the field. The kit used to perform the analysis on-site is easy to use and, in our opinion, could allow an increase of analysis of different matrices by performing rapid analysis and being free from having to use a laboratory instruments.

Certainly the duration of the film is currently the major limit of the proposed procedure, the idea is to find better conditions to improve the durability of the film. Surely in the future we will test further changes in the surface of the GCE. In our opinion, the technique is innovative because it allows one to do direct analysis even in remote areas, in order to monitor iron concentrations directly, avoiding sampling and sample transport that can be difficult and challenging in some cases.

We mention a couple of examples hereafter:

- we have been working for years in environmental monitoring in remote areas, namely Antarctica and the Arctic: we believe that our kit will allow the analysis of water (and not only) directly in the laboratories on-site avoiding the transport of the sample that is expensive and often very difficult, especially if you want to carry out speciation studies.

- the final goal of our research study will be to apply the methodology on biological samples (saliva, serum, etc.): also in this case the possibility to do analysis on site soon after sampling, in order to avoid changes in element speciation, and with an easily usable method will, in our opinion, be fundamental.

A sentence has been added to the manuscript:

Line 274“The primary objective is to identify optimal conditions that enhance the film's longev-ity. Subsequently, it is anticipated that future investigations will encompass additional surface modifications of the Glassy Carbon Electrode (GCE) to explore further enhancements.” and line 280“The innovative nature of this technique concerns in its capacity to facilitate direct an-alytical assessments in remote geographic areas, enabling the real-time monitoring of iron concentrations while avoid the logistical complexities associated with sample col-lection and transportation, particularly in challenging environmental contexts.”

Another thing: what are the advantages of the proposed procedure in comparison to the others methodologies, known in literature?

The biggest advantage, as mentioned above, is the ability to perform routine analysis anywhere. Moreover, expensive instrumentation is required to performed iron speciation studies in laboratory, if compared with our proposed procedure.

Following remarks should be also taken into account:

  1. The novelty of the manuscript is rather limited. The use of Sb should be explained, why the Authors decided to modify the electrode surface together with Sb and Bi??

Thanks for the question. Bismuth and antimony adopter together better improve the electrochemical performance of glassy carbon electrodes (GCE) than adopted alone. These metals in combination can provide a more active surface for electrochemical reactions, leading to increased sensitivity and selectivity in various electrochemical sensing and detection applications. Bismuth and antimony-modified electrodes often exhibit low background currents, which are essential for improving the signal-to-noise ratio in electrochemical measurements. This feature is especially important in trace analysis and sensing applications. Bismuth and antimony-modified electrodes have a high affinity for specific metal ions. This selectivity is particularly useful in environmental monitoring, where the presence of certain heavy metal ions needs to be detected accurately. Bismuth and antimony-modified electrodes can enhance the performance and selectivity of these portable systems for make on-site analysis.

Maybe some kind of comparison, of the developed sensor should be performed, for example bare electrode vs Bi modified electrode vs Sb/Bi modified electrode. And based on such results it can be concluded does this research make sense.

In the preliminary studies and in previous studies we tried bare-GCE, modified only with bismuth (BiFE-GCE) and modified with both (SbBiFE-GCE). We valued the percentage of recovery on 5 µg L-1 of iron (III).

At this concentration the bare-GCE does not allow the determination of this concentration, the BiFE-GCE permitted to obtain a recovery of 85.6%, while the SbBiFE-GCE, as shown in table 3, lead a recovery of 95.4%. For the SbBiFE-GCE as shown in table 2 (now table 1), different concentrations and depositions times of antimony and bismuth have been tested to have the best performance.

  1. Page 5, what is the linear range of the proposed procedure?

The linear range of the proposed procedure is 5-20 µg L-1.

  1. What about Figure 2? Is it not mentioned in the text, what is the conclusion about these data??

Figure 2 is concerned with demonstrating that the technique is selective for the determination of iron (III), three additions of 5 µg L-1 of iron (II) have been made, no peak appears as a consequence of these additions. Then, an addition of iron (III) was made and the corresponding peak appears.

The following sentence has been added: "In figure 2 the voltammogram of three standard addition of iron (II) and a subsequent one of iron (III) demonstrating the selectivity of the method for iron (III)".

How the peak currents were measured?

Thanks for the question, the height of the peaks is taken automatically by the software (PS trace 5.9). We checked that the correct baseline was selected, and modified it if necessary.

We have added this detail in the text “The height of peaks were always taken in “automatic mode” by the software (PS trace 5.9)”.

  1. If the Authors mentioned that some of analyses were performed outside of the lab, then it is interesting to look at these data, if the signals differ somehow. Then Authors should add voltammograms for on situ analyses.

The voltammogram of a measurement carried out in the field was added (figure 6), the only criticality found was a slight increase in the background currents, which however have not affected the goodness of the analysis.

The following sentence has been added: “ Figure 6 shows a voltammogram of an on-site  analysis of lake water.

Figure 6 – Voltammograms recorded for lake water on-site analysis.

List of changes

All the revisions are referred to the file “molecules-2671518_Highlighted”, in this file the added sentences are in red, while the deleted ones are blue and crossed out.

Reviewer 2 Report

Comments and Suggestions for Authors

1) Authors must report in the abstract the main results found in the research.

2) The authors talk about the perspective of using the electrode modified with Bi and Sb for biological samples, such as blood, I ask: how will they solve the problem of the matrix effect, as blood plasma is as complex a matrix as lake water?

I suggest that actors write in the Results and discussion section a way to solve the matrix effect problem.

Author Response

1) Authors must report in the abstract the main results found in the research.

Thanks for the observation, we have modified the last part of the abstract with: “, then a Standard Reference Material (SRM) 1643f - Trace Elements in Water (iron content: 93.44 ± 0.78 µg L-1) was used for validation measurements in real matrix, the accuracy of the technique was found to be excellent, since we obtain a recovery of 103.16%. The procedure was finally applied to real samples (tap, lake and sea water) and the results obtained were compared by Inductively Coupled Plasma Optical Emission Spectroscopy (ICP-OES). The amount of iron found was 207.8 ± 6.6 µg L-1 for tap water using voltammetry and 200.9 ± 1.5 µg L-1 with ICP-OES. For lake water, 171.7 ± 3.8 µg L-1, 169.8 ± 4.1 µg L-1 and 187.5 ±- 5.7 µg L-1 were found by voltammetry in lab, on-site and ICP-OES respectively. The results obtained demonstrated the excellent applicability of the proposed on-site voltammetric procedure for the determination of iron and its speciation in water.

2) The authors talk about the perspective of using the electrode modified with Bi and Sb for biological samples, such as blood, I ask: how will they solve the problem of the matrix effect, as blood plasma is as complex a matrix as lake water?

I suggest that actors write in the Results and discussion section a way to solve the matrix effect problem.

We thank the Reviewer for the right observation.

In order to decrease the matrix effect that we will surely have with biological samples, in the first place the idea is to dilute the samples at least 1:10. This would however allow us to determine iron, as in previous studies we have seen that the concentration of iron in biological samples of patients not suffering from neurodegenerative diseases the total iron concentrations were about 30/40 µg L-1 (analysis made with atomic absorption spectroscopy).

Other approaches that it is possible to adopt are for example centrifuge the sample to separate solid particles or precipitates from the liquid phase, adjust the pH of the sample to a suitable level to enhance iron complex formation while minimizing interference. This can often be achieved with a buffer solution (as the one that we used in our method). Unfortunately, these treatments would hinder the measurement of iron speciation, and only total iron would be detected.

Also for biological matrices the method of the standard addition will be used to quantify the analyte. Adding known amounts of a standard iron solution to the sample and measuring the current response. This approach can help account for matrix effects and accurately determine the iron concentration.

We have modified part of the conclusions and integrated the paragraph in the following way in the part of results and discussion: “Future development of this study will be testing the applicability of the procedure also to the analysis of biological samples (such as plasma or serum). Even more challenging is the detection of iron in the CerebroSpinal Fluid (CSF), ranging from about 30 in controls and 70 μg L-1 in AD patients [39]. The determination of iron in CSF samples is carried out by means of Graphite Furnace Atomic Absorption Spectrometer (GF-AAS), which is a very precise but time requiring method. For the voltammetric analysis of CSF, matrix effects have to be removed. The following approaches may be used: centrifuge the sample to separate solid particles or precipitates from the liquid phase, adjust the pH of the sample and dilute the samples with the supporting electrolyte at least 1:10.”

List of changes

All the revisions are referred to the file “molecules-2671518_Highlighted”, in this file the added sentences are in red, while the deleted ones are blue and crossed out.

Reviewer 3 Report

Comments and Suggestions for Authors

Table 1 should be deleted and the data should be included within the text.

Calibration curve is missing.

Reproducibility study is missing.

Interference study was not performed.

Standard addition method should be performed at several concentration ranges.

Introduction should be enriched with recent publications.

Conclusions should be concise and updated with newly performed studies.

Author Response

Table 1 should be deleted and the data should be included within the text.

Thank you for the suggestion, table 1 has been deleted, together with the sentence “Table 1 shows the SWAdSV parameters optimised for the determination of iron (III) and which will be used in all subsequent tests”. The following sentence has been added “Parameters optimized for iron (III) determination include a deposition at -0.4V for 180 s and a subsequent scan from -0.3 V to -0.9 with a frequency of 10 Hz and an amplitude of 0.025 V.”

Calibration curve is missing.

The standard addition method was used to quantify concentrations, so an external calibration curve was not necessary.

Reproducibility study is missing.

Thanks for the observations, reproducibility studies were carried out under different conditions intracell, intercell, in laboratory and on-site. The following sentences has been added: “Reproducibility studies were performed intracell, intercell, in laboratory and in field with relative % RSD of 2.8%, 3.0%, 4.1% and 3.9%.”

Interference study was not performed.

Thanks for the observation.

No interference study has been carried out because the reduction potential studied is specific to iron. Moreover, by analyzing the Standard Reference Material (SRM) 1643f Trace Elements in Water, which contains 29 different analytes, no interference effects were found in our method.

Standard addition method should be performed at several concentration ranges.

The standard addition method was used to determine element concentrations. We focused on this concentration range as it was possible to dilute the samples.

Introduction should be enriched with recent publications.

Thank you for the suggestion.

We have added some recent papers in the introduction, in particular:

14]       S. L. Baringer, I. A. Simpson, e J. R. Connor, «Brain iron acquisition: An overview of homeostatic regulation and disease dysregulation», J. Neurochem., 165, 5, 625–642, 2023, doi: 10.1111/jnc.15819.

[15]      Y. Xu, X. Huang, X. Geng, e F. Wang, «Meta-analysis of iron metabolism markers levels of Parkinson’s disease patients determined by fluid and MRI measurements», J. Trace Elem. Med. Biol., 78, 127190, 2023, doi: 10.1016/j.jtemb.2023.127190.

[32]      R. Ferreira, J. Chaar, M. Baldan, e N. Braga, «Simultaneous voltammetric detection of Fe3+, Cu2+, Zn2+, Pb2+ e Cd2+ in fuel ethanol using anodic stripping voltammetry and boron-doped diamond electrodes», Fuel, 291, 120104, 2021, doi: 10.1016/j.fuel.2020.120104.

[37]      X. Ma, J. Yu, L. Wei, Q. Zhao, L. Ren, e Z. Hu, «Electrochemical sensor based on N-CQDs/AgNPs/β-CD nanomaterials: Application to simultaneous selective determination of Fe(â…¡) and Fe(â…¢) irons released from iron supplement in simulated gastric fluid», Talanta, 253, 123959, 2023, doi: 10.1016/j.talanta.2022.123959.

We have modified the introduction accordingly, adding some sentences:

  • In a recent paper Baringer et al. [14] emphasize the fundamental roles that iron plays for human neurological health, in this way…
  • .. ], boron-doped diamond (BDD) electrodes [32] …
  • In recent years even more complex electrode modifications have been tested for simultaneous iron (II) and iron (III) detection using eg nitrogen-doped carbon quantum dots silver nanoparticles β-cyclodextrin nanomaterials (N-CQDs/AgNPs/β-CD) [37].

Conclusions should be concise and updated with newly performed studies.

Thank you for your suggestion, the conclusions have been reformulated as follows:

The proposed method has been shown to be excellent for the determination of iron (III) in water, using a method also applicable in field. The results obtained, the precision and accuracy of the optimized procedure make it possible to hypothesize its application in different areas. A future development of the method can be to increase the stability of the Bi/Sb, this reducing the frequency of its regeneration. Anyway, the time required for a complete voltammetric analysis, including film deposition, is 30 minutes: this value is acceptable, and a throughput of 25 samples for 8 hours of operation is expected. The method is suitable also for speciation studies, considering that iron (II) is not detected using the proposed experimental conditions. This can allow a potential increase in water analyses in case of iron contamination, or to investigate the role of this metal in biogeochemical cycles, even directly on site. This possibility will enable researchers to have rapid and immediate monitoring of iron concentration, testing it suitability as a possible marker for early detection of neurodegenerative diseases.

List of changes

All the revisions are referred to the file “molecules-2671518_Highlighted”, in this file the added sentences are in red, while the deleted ones are blue and crossed out.

Round 2

Reviewer 1 Report

Comments and Suggestions for Authors

Thank you for all the answers to my questions. I accept the article in its current form.

Reviewer 3 Report

Comments and Suggestions for Authors

The Authors have performed extended work to correct the manuscript and answer requested comments.